# Differential Effects of Canonical Androgens and 11-Ketotestosterone on Reproductive Phenotypes and Folliculogenesis in Mouse Model of PCOS

**DOI:** 10.3390/biomedicines13051077

**Published:** 2025-04-29

**Authors:** Yi-Ru Tsai, Yen-Nung Liao, Cheng-Ju Tsai, Yu-Ang Lee, Shih-Min Hsia, Kuo-Chung Lan, Hong-Yo Kang

**Affiliations:** 1Graduate Institute of Clinical Medical Sciences, College of Medicine, Chang Gung University, Taoyuan 333, Taiwan; 2An-Ten Obstetrics and Gynecology Clinic, Kaohsiung 802, Taiwan; 3Department of Chinese Medicine, Kaohsiung Chang Gung Memorial Hospital, College of Medicine, Chang Gung University, Kaohsiung 833, Taiwan; 4Graduate Institute of Metabolism and Obesity Sciences, School of Food Safety, College of Nutrition, Taipei Medical University, Taipei 110, Taiwan; 5Center for Hormone and Reproductive Medicine Research, Department of Obstetrics and Gynecology, Kaohsiung Chang Gung Memorial Hospital, College of Medicine, Chang Gung University, Kaohsiung 833, Taiwan; 6Division of Endocrinology and Metabolism, Department of Internal Medicine, Kaohsiung Chang Gung Memorial Hospital, College of Medicine, Chang Gung University, Kaohsiung 833, Taiwan; 7Department of Biological Science, National Sun Yat-sen University, Kaohsiung 804, Taiwan

**Keywords:** polycystic ovary syndrome, hyperandrogenism, androgen receptor, 11-ketotestosterone

## Abstract

**Background**: Polycystic ovary syndrome (PCOS) is a common female endocrine disorder characterized by hyperandrogenism, ovulatory dysfunction, and polycystic ovarian morphology. While canonical androgens like testosterone (T) and dihydrotestosterone (DHT) are well studied in PCOS pathophysiology, the role of 11-ketotestosterone (11KT) remains unclear. This study investigates the differential effects of these androgens on folliculogenesis, ovulation, and steroidogenesis using in vivo and in vitro models. **Methods**: Four-week-old female C57BL/6 mice received T, DHT, or 11KT for six weeks. The assessments included body weight, estrous cyclicity, serum hormone profiles, ovarian histology, and follicle classification. In parallel, large preantral follicles were cultured with each androgen to evaluate follicle growth, antrum formation, and ovulation capacity. Androgen receptor (AR) signaling and steroidogenic function were analyzed using western blotting, RT-qPCR, and luciferase reporter assays. **Results**: The DHT-treated mice exhibited increased weight gain, whereas 11KT-treated mice showed reduced weight gain. T and DHT disrupted the estrous cycle, while 11KT prolonged diestrus. All androgen treatments led to ovarian morphological changes, including follicular arrest and cystic features. In vitro, all androgens enhanced follicle growth, but only T and DHT inhibited ovulation. The AR expression was elevated across all androgen-treated groups, but only DHT significantly activated AR and CYP19A1 promoters. **Conclusions**: 11KT induces a distinct and milder PCOS-like phenotype compared to classical androgens, promoting follicle growth with minimal impact on ovulation or steroidogenic disruption. These findings underscore the heterogeneity of PCOS and suggest that different androgen profiles may drive diverse clinical phenotypes. By elucidating the distinct roles of different androgens, this may lead to better stratification of PCOS phenotypes based on predominant androgen types for more precise diagnosis and individualized management.

## 1. Introduction

Polycystic Ovary Syndrome (PCOS) is one of the most common gynecological diseases and occurs in 6 to 10% of women of reproductive age [1]. Based on the Rotterdam criteria, PCOS is subdivided into four distinct phenotypes that differ in severity and clinical presentation, from the classic form characterized by hyperandrogenism and anovulation to milder forms with only polycystic ovarian morphology [2]. As a hallmark of PCOS, hyperandrogenism contributes to clinical manifestations such as hirsutism, acne, and alopecia, while also disrupting normal folliculogenesis and ovarian function [3]. Beyond reproductive issues, most women with PCOS also face metabolic challenges such as obesity, dyslipidemia, insulin resistance, and a heightened risk of type 2 diabetes mellitus and cardiovascular disease [1]. Growing evidence indicates that hyperandrogenism may significantly contribute to these metabolic changes by worsening insulin resistance and impairing adipocyte function, thereby creating a vicious cycle between androgen excess and metabolic dysregulation [4].

Androgens exert biological effects via nuclear androgen receptor (AR) genomic signaling and non-genomic signaling involving the MAPK-ERK and AKT pathways [4]. The androgen actions via the AR play a crucial role in PCOS development [5]. Additionally, prenatal androgen exposure has been implicated in the developmental origins of PCOS [6]. Walters et al. indicated that genomic AR signaling is crucial for PCOS induction via prenatal hyperandrogenization and that even partial AR haplosufficiency can prevent PCOS-like traits in adulthood, suggesting a dose-dependent effect of androgen signaling [6]. AR signaling in neurons is a key driver of PCOS traits and is necessary for the development of ovulatory dysfunction, polycystic ovarian morphology, and metabolic abnormalities in the dihydrotestosterone (DHT)-induced PCOS mouse model [7], indicating that extraovarian AR-mediated mechanisms contribute significantly to PCOS pathology, providing a potential therapeutic target. This also clarifies the distinct pathways through which androgens exert their effects. Moreover, metabolic disturbances are primarily driven by direct AR action, whereas reproductive dysfunction involves both the AR and estrogen receptor (ER) signaling in AR knockout (ARKO) mice [8]. These findings highlight the importance of considering phenotype-specific mechanisms when designing therapeutic interventions.

Rodent models have provided key insights into the pathophysiology of the syndrome, particularly regarding androgen-mediated mechanisms [7]. Mouse models have demonstrated that direct AR signaling is critical in driving the reproductive and metabolic phenotypes associated with the disorder [8], and have been instrumental in characterizing its metabolic features. Among these PCOS models, the DHT-induced model replicates key metabolic abnormalities, including increased adiposity, insulin resistance, and dyslipidemia [9] while providing crucial insights into the association between reproductive and metabolic dysfunction. Similarly, Walters et al. conducted a comprehensive evaluation of multiple hyperandrogenic mouse models and identified long-term DHT treatment as the most effective in reproducing both reproductive and metabolic traits [10]. These studies indicate that targeting androgen–AR improves chronic anovulation and menstrual irregularities in PCOS animal models [11], suggesting it as a potential therapy for hyperandrogenic condition [12].

As an accepted biomarker for androgen excess in PCOS, serum T is converted to DHT due to systemic upregulation of 5α-reductase activity in affected women [13]. T precursor androstenedione also has been shown to be a more sensitive marker of PCOS-related androgen excess [14]. Notably, a novel class of active androgens, 11-oxygenated androgens, is significantly elevated in women with PCOS, with or without obesity, account for a greater proportion of total circulating androgens, and are better predictors of PCOS [15,16,17] than classical androgens. Furthermore, in a previous study [18], the cases of PCOS were characrized with elevated 11-oxygenated androgens but a normal level of classical androgens. It is clear that androgen excess in women with PCOS is heterogeneous and complicated, highlighting the need for further research on 11-oxygenated androgens. One of the 11-oxygenated androgens is 11-ketotestosterone (11KT), which was first characterized as a major male hormone in teleost and evolutionally conserved in mammals [19,20,21,22]. It is also an active nonaromatizable androgen in human gonads and adrenal glands, necessary for activating AR-mediated transactivation [19,23]. Testicular leydig cells and ovarian theca cells express the steroidogenesis enzymes CYP11B1 and HSD11B2 to produce 11KT from T [19]. Although, it was reported that both reproductive-age and menopausal women maintain similar levels of T and 11KT [20], 11KT circulates at significantly higher concentrations than T, and its levels are significantly elevated in women with PCOS [15]. These findings suggest that 11KT may play a crucial role in PCOS pathophysiology and could potentially serve as a more sensitive biomarker for hyperandrogenism in women. These insights open novel avenues for developing an 11KT-induced PCOS mouse model to explore the androgen excess effects of 11-oxygenated androgens in PCOS.

## 2. Materials and Methods

### 2.1. Animals and Study Design

C57BL/6NCrlBltw female mice were obtained from BioLASCO (BioLASCO Taiwan Co., Ltd., Taiwan) and housed in a pathogen-free facility at Kaohsiung Chang Gung Memorial Hospital. The mice were maintained under controlled conditions with a 12-h light/dark cycle, a temperature of 22 ± 1 °C, and a relative humidity of 50 ± 5%. They were provided with free access to a cereal-based pelleted diet (1324, Altromin, Germany) and autoclaved, reverse osmosis-filtered water. The animals were housed in individually ventilated polycarbonate cages (Tecniplast, Italy) with a floor area of approximately 500 cm^2^ per cage, with 4–5 mice per cage in accordance with the institutional animal welfare guidelines. Each cage was provided with dust-free spruce woodchip bedding in cubic granulate form (FS14 and 3-4S, SAFE, Germany), which was replaced twice weekly. All experimental procedures were reviewed and approved by the Institutional Animal Care and Use Committee of Chang Gung Memorial Hospital. To induce PCOS-like traits, 4-week-old female mice were subcutaneously injected once daily with androgen dissolved in polyethylene glycol 400 (PEG400, P3265, Sigma-Aldrich, USA). The PEG400 injection alone served as the vehicle (VE) group. After six weeks of drug administration, the mice (T, n = 21; DHT, n = 19; 10 mg/kg/body weight 11KT, n = 20; and 25 mg/kg/body weight 11KT, n = 14, Ctrl, n = 19) were sacrificed to collect the ovaries and serum. T (86500, Sigma-Aldrich, USA, 10 mg/kg/body weight), DHT (A8380, Sigma-Aldrich, USA, 15 mg/kg/body weight), and two 11KT dosages (Yu Shing Tang Taiwan Co. Ltd., Taiwan, 10 and 25 mg/kg/body weight) were used to induce PCOS-like traits in mice. The dosage of androgen required to induce PCOS was based on previous studies [9,24]. The mice were routinely monitored to ensure their microbiological status, general health, welfare, and adequate environmental enrichment such as shelters and tunnels. At the end of the treatment period, all mice were humanely euthanized by CO_2_ inhalation. The group sizes were determined based on power calculations to ensure sufficient statistical power (α = 0.05) to detect biologically meaningful differences in outcome measures.

### 2.2. Assessment of the Estrous Cycle

After vaginal cells were collected using 0.9% sterile saline, their morphology was observed with toluidine blue solution (ScyTek Laboratory, Logan, UT, USA) under the microscope, as previously described [25]. The four phases of the estrous cycle (proestrous, estrous, metaestrous, and diestrous) were determined daily by vaginal smear for 10 days after a 4.5-week injection.

### 2.3. Mouse Ovary Collection and Follicle Classification

Dissected ovaries were weighed, dehydrated, and paraffin-embedded at the pathological examination department of Kaohsiung Chang Gung Memorial Hospital. The 5 μm serial tissue sections of paraffin-embedded mouse ovaries were deparaffinized and rehydrated from graded ethanol (100%, 95%, 70%) before eosin (HT110132, Sigma-Aldrich, USA) and hematoxylin (HX690812, Merck, Germary) staining. Follicle classification was based on a previous study [10]. The number and percentage of primordial follicles (oocyte with one layer of flattened granulosa cells), primary follicles (oocyte with one layer of cuboidal granulosa cells), small preantral follicles (oocyte with 2 layers of cuboidal granulosa cells), large preantral follicles (oocyte with >2 layers of cuboidal granulosa cells), small antral (oocyte surrounded with >5 layers of granulosa cells, and 1 or 2 small areas of follicular fluid), large antral follicles (contained 1 large antral cavity), corpora lutea (a mass from the ruptured graafian follicle), atretic preantral follicles, atretic antral follicles (contained a degenerate oocyte or >10% loss of the granulosa cells or contained an attenuated granulosa cell layer, dispersed theca cell layer, and an oocyte lacking connection with the granulosa cells) were counted on serial sections throughout each ovary. For all large antral follicles, the thickness of the granulosa cell layer and theca layer was measured using Image J v.1.50b software (RRID:SCR_003070).

### 2.4. Hormone Assays

The blood samples from the mice were collected through cardiac puncture. The mouse serum was stored at −80 °C. The serum hormone levels of LH (ab235648, Abcam, UK), 17β-estradiol (ADI-900-174, Enzo, USA), T (ab108666, Abcam, UK), DHT (E-EL-0031, Elabscience, China), and 11KT (582751, CAYMAN, USA) were assessed via ELISA.

### 2.5. Mouse Follicle Isolation and In Vitro Culture

Ovaries from 3-week-old C57BL/6 female mice were dissected and used for follicle isolation. Large preantral follicles (LPAFs) measuring 110–170 μm in diameter, exhibiting intact granulosa cell layers, a round oocyte, and a well-defined basement membrane, were carefully isolated using acupuncture needles under a stereomicroscope, as described previously [26].

The follicles were individually cultivated in 96-well plates (1 follicle/well), each well containing α-minimum essential medium (12571063, Gibco, USA) containing 10% of charcoal/dextran-treated fetal bovine serum (CD-FBS, 12676029, Gibco, USA), recombinant human FSH (10 ng/mL, Gonal-F, Merck, Germary), insulin (10 mg/L), transferrin (5.5 mg/L), sodium selenite (6.7 μg/L, Corning, USA), penicillin (100 units/mL), and streptomycin (100 mg/L; 15240096, Gibco, USA). The plates were maintained at 37 °C in a humidified incubator with 5% CO_2_, with half of the medium replaced every 2 days.

### 2.6. Isolation of Mouse Primary Granulosa Cells and Cell Culture

The 3-week-old mouse ovaries were washed with phosphate buffer saline (PBS). Then, minced tissues were digested with collagenase (1 mg/mL; S1746501, Nordmark, Germany) in 0.25% trypsin (15050065, Gibco, USA) and centrifuged at 1000 rpm and 37 °C for 60 min. Mouse primary granulosa cells (MPGCs) were separated by filtration using a 40 μm pore size nylon mesh. Filtered MPGCs were allowed to attach on a dish overnight, and then blood cells and tissue debris were washed away with PBS. MPGCs and KGN (RCB1154, RIKEN BRC Cell bank, Japan, RRID: CVCL_0375) were cultured in DMEM/F12 medium with 10% FBS (10437028, Gibco, USA) in a humidified atmosphere with 5% CO_2_ at 37 °C.

### 2.7. Western Blotting and RT-qPCR Analysis

The total protein lysates subjected to Western blotting and Total RNA were subjected to RT-qPCR, as previously described [27]. Antibodies against AR (ab133273, Abcam, USA, RRID: AB_11156085) and GAPDH (MA5-15738, Invitrogen, USA, RRID: AB_10977387) were used for Western blotting. We performed RT-PCR in Fast SYBR green master mix kit (4385612, Applied Biosystems, USA). The sequences were analyzed using an ABI 7500 Fast sequence detection system (Waltham, MA, USA). The transcripts of *β-actin* served as the endogenous RNA control and normalized each sample. The relative transcription levels of genes were calculated using 2^−ΔΔCT^ methods. The mouse qPCR primers sequence was as follows: *β-actin* F: AGGCCAACCGTGAAAAGATG, *β-actin* R: TGTGGTACGACCAGAGGCATAC, *FSHR* F: CCTTGCTCCTGGTCTCCTTG, *FSHR* R: CTCGGTCACCTTGCTATCTTG, *LHCGR* F: GATGCACAGTGGCACCTTC, *LHCGR* R: TCAGCGTGGCAACCAGTAG, *CYP19A1* F: GAGAGTTCATGAGAGTCTGGATCA, *CYP19A1* R: CATGGAACATGCTTGAGGACT, *STAR* F: CCGGAGCAGAGTGGTGTCA, *STAR* R: CAGTGGATGAAGCACCATGC. The secondary antibody was used for incubation for 1 h, and immunodetection was conducted using ECL (PerkinElmer^TM^, USA).

### 2.8. Luciferase Assay

KGN cells were transiently transfected with the mouse mammary tumor virus (MMTV)–luciferase or *CYP19A1*-promoter II-589bp-luciferase construct via Lipofectamine™ 2000 Transfection Reagent (11668019, Thermofisher, USA) according to the manufacturer’s instructions. Cells were co-transfected with the Renilla luciferase (SV40) plasmid as the internal control. After transfection, the cells were treated with DHT or 11KT for 48 h. The dual-luciferase reporter assay system (E1910, Promega, USA) was applied to determine the ratio of the firefly luciferase activity to the Renilla luciferase activity in the samples.

### 2.9. ELISAs for 17β-Estradiol Quantification

Seeded 5 × 10^4^/cm^2^ MPGCs and 10^4^/cm^2^ KGN were cultivated in phenol red-free DMEM/F12 medium (21041025, Gibco, USA) with 10% CD-FBS and 1 nM T. Cells and LPAFs (8 follicles/mouse/group) from mice treated with 1 μM T, DHT, and 11KT for 48 h to generate different hyperandrogenic environment. The collected culture media were used to measure the estrogen levels using the 17β-estradiol ELISA kit (ADI-901-008, Enzo, USA).

### 2.10. Statistical Analysis

Independent experiments were performed in triplicate. The data were presented as mean ± standard error of the mean (SEM) via GraphPad Prism 6 (RRID:SCR_002798). For parametric tests, the statistical significance between the two groups was determined by a two-tailed Student’s *t*-test and three or more groups was evaluated by a one-way ANOVA with Bonferroni’s post-hoc test. The *p*-value of less than 0.05 was considered statistically significant.

## 3. Results

### 3.1. Establishment and Comparison of Hyperandrogenized Mouse Models for Different Traits of PCOS

While androgen excess is the major symptom of PCOS and both classical and 11-oxygenated androgens levels are increased in patients with PCOS [23], it remains unclear whether the excess of these androgens contributes to the impairment of follicle function and estrogen production in PCOS. To compare the differential effects of androgens on the phenotypes of PCOS mouse models, we injected T, DHT, and 11KT into 4-week-old mice subcutaneously for 6 weeks to generate PCOS-like mouse models (Figure 1A) and assessed 6-week weight gain, estrous cycle phase, and hormone levels. Compared with vehicle-treated mice, which exhibited normal estrous cycles, the T- and DHT-treated mice displayed estrous cycle arrest in the diestrous phase, and 11KT-treated mice showed estrous cycle delayed estrous cycle with a prolonged diestrous phase (Figure 1B–D). Weight gain was increased in DHT-treated mice but decreased in 11KT-treated mice compared with that of vehicle-treated mice (Figure 1E).

To further investigate the effect of hyperandrogenism on the female reproduction system in mice, we measured ovary weight, conducted histological analysis and performed follicle counting. Since the ovary weight (Figure 2A) and the distribution of total and atretic follicles population (Figure 2B,C) were similar for the two doses of 11KT-treated mice, we selected the 10 mg/kg dose as the representative for comparison with other androgens. The mice treated with T, DHT, or 11KT exhibited reduced ovarian weight and size compared with vehicle-treated mice (Figure 3A,B). The ovarian histology and follicle classification revealed that androgen-treated mice exhibited multicystic ovaries (Figure 3C) with thicker theca cell layers and thinner granulosa cell layers (Figure 3D,E). Additionally, the proportion of corpus luteum was reduced, and that of large antral follicles was elevated in T-, DHT- and 11KT-treated mouse ovaries compared with those in vehicle-treated mice (Figure 3F). Next, T-, DHT- and 11KT-treated mice showed significant increases in both atretic preantral and antral follicles, but the atretic antral follicles in 11KT-treated mice were less increased than other androgen-treated mice compared with the vehicle-treated mice (Figure 3G). The serum analysis revealed elevated LH levels alongside corresponding androgen increases in all androgen-treated mice (Table 1). Moreover, both estrone (E1) and 17β-estradiol (E2) levels in serum were notably increased in T- and DHT-treated mice (Table 1).

### 3.2. Promotion of Follicle Growth Caused by High Androgen Level and Its Negative Impact on Survival Rate and the Function of Ovulation in In-Vitro Follicle Culture System

Androgen excess is a characteristic of PCOS, and both classical and 11-oxygenated androgen are overproduced in PCOS patients. However, whether excess 11-oxygenated androgens contribute to folliculogenesis and ovulatory dysfunction in PCOS remains unclear. To explore the different effect between classical and 11-oxygenated androgen on folliculogensis, we collected 110–170 μm follicles, seeded one follicle one well in 96-well plate, and supplemented the culture medium with T, DHT, or 11KT to simulate a hyperandrogenic environment. After 7 days of in vitro culture, the follicle diameter (Figure 4A,B) and antrum formation rate (Figure 4C,D) were increased in T, DHT and 11KT treated follicles compared to follicles without androgen treatment. Compared with the vehicle group, T and DHT exposure inhibited hCG-induced in vitro ovulation, whereas 11KT had no significant impact on ovulation, but excess 11KT showed no impact on hCG-induced ovulation (Figure 4E,F). These findings suggested that 11KT, a comparatively mild androgen, promotes follicle growth and induces cyst-like follicle formation, consistent with our observations in the 11KT-treated mouse model. However, 11KT had a limited effect on ovulation and luteinization processes in mature follicles.

### 3.3. Distinct Effects of 11KT on AR Expression and Downstream Signaling in PCOS Models

To further characterize different genomic or non-genomic androgen actions on the folliculogenesis and steroidogenesis in hyperandrogenized models, we performed western blotting and RT-qPCR analysis. The AR protein expression in minced ovary samples (Figure 5A) and MPGCs (Figure 5B) were increased in androgen treated groups compared to the vehicle group. We further analyzed the expression of genes related to folliculogenesis, including follicle-stimulating hormone receptor (*FSHR*) and LH/choriogonadotropin receptor (*LHCGR*), as well as genes involved in steroidogenesis, namely Cytochrome P450 Family 19 Subfamily A Member 1 (*CYP19A1*) and Steroidogenic Acute Regulatory Protein (*STAR*), in the ovary (Figure 5C) and MPGCs (Figure 5D). We found that *FSHR*, *LHCGR* and *CYP19A1* were upregulated and *STAR* was downregulated under classical androgen treatment (especially DHT). In contrast, no significant changes were observed in the AR downstream target gene expression in 11KT-treated ovaries and MPGCs. Similarly, luciferase assay demonstrated that the transcriptional activity of AR (Figure 5E) and CYP19A1 promoter II (Figure 5F) was activated by excess DHT, but not by 11KT. In parallel, we collected an in vitro culture medium from androgen-treated mouse follicles, MPGCs and KGN treated with different androgens in vitro to compare their capacity for estrogen production. E2 concentrations were elevated in the culture medium of mouse follicles, MPGCs, and KGN cells treated with T and DHT, but remained unchanged in the 11KT-treated groups (Figure 5G).

## 4. Discussion

PCOS is a complex endocrine disorder reproductive, hormonal, metabolic, and psychological symptom, impacting reproductive-age women. Hence, no ideal animal PCOS model encompassing the entire spectrum of PCOS manifestations and mimicking all reproductive, hormonal, and metabolic traits exists yet. Among the methods of inducing hyperandrogenic PCOS in animal models, androgenic steroids (DHEA, DHT and T) and letrozole (aromatase inhibitor) are the most frequently used drugs to recapitulate both reproductive and metabolic phenotypes, including elevated LH level, anovulation, increased body fat, and insulin resistance [28]. As the majority of circulating androgens in women with PCOS, with significant correlation to metabolic risk markers, 11-oxygenated androgens have not been used to investigate the underlying mechanism of PCOS. In this study, we aimed to compare the varying effects of androgens and to explore the molecular mechanisms by which hyperandrogenism impairs reproductive function in PCOS. By utilizing in vivo and in vitro models, we demonstrated that excess androgens differentially regulate steroidogenesis and folliculogenesis in PCOS. Notably, the activation of androgen–AR signaling by various androgens plays a crucial role in PCOS pathogenesis.

Both aromatizable androgen and non-aromatizable androgen were able to induce reproductive PCOS traits, including acyclicity, anovulation, and polycystic ovarian morphology in mice. This confirmed that androgen-driven mechanisms are central to PCOS pathogenesis [10,29]. Walters et al. reinforced the importance of direct AR-mediated androgen actions, while also highlighting a potential indirect role for ER signaling in the development of reproductive features of PCOS [8], suggesting that, in the absence of AR, aromatizable androgens may act via ER pathways to induce reproductive dysfunction, a hypothesis supported by clinical evidence that anti-estrogens such as letrozole and clomiphene are effective ovulation induction therapies in PCOS [30,31,32]. The neuroendocrine axis plays a crucial role in reproductive function, and emerging evidence suggested that dysregulated estrogen signaling, in addition to AR-driven mechanisms, may contribute to PCOS pathogenesis. The disruptions in hypothalamic ERα signaling have been linked to aberrant LH secretion and polycystic ovarian morphology [33], aligning with prior work that identified AR-mediated neuroendocrine dysfunction in PCOS models [7,34]. This raises the possibility that excess androgens, through both AR and ER pathways, may contribute to the neuroendocrine abnormalities observed in PCOS. The heterogeneity of PCOS phenotypes may be explained by variations in these underlying steroids signaling pathways. AR-driven mechanisms are essential for the reproductive dysfunction seen in hyperandrogenic PCOS phenotypes, whereas ER-mediated pathways may contribute to the reproductive disturbances observed in non-hyperandrogenic PCOS [2,35]. This distinction is clinically relevant, as it suggests that AR antagonists may be beneficial for hyperandrogenic PCOS cases, while ER modulators could be more appropriate for non-hyperandrogenic PCOS patients. Taken together, these findings underscore the complexity of androgen-driven reproductive dysfunction in PCOS. While direct AR-mediated actions remain central to the pathogenesis of reproductive disturbances, indirect effects via ER signaling may also play a role. Future studies incorporating ER antagonists, along with detailed characterization of intracranial androgen metabolism, will be critical in further delineating the specific pathways through which androgens drive reproductive dysfunction in PCOS.

Serum concentrations of 11-oxygenated androgens were reported to be increased in patients with PCOS, independent of BMI [15]. The level of 11KT was significantly higher than that of T in women with PCOS [15]. Although the biological roles of 11-oxygenated androgens contribute to PCOS pathophysiology, this finding remains insufficiently investigated. Here, we generated a 11KT-induced PCOS mouse model and compared the difference between this model and mouse models of PCOS induced by classical androgens. Consistent with previous studies [36,37], T- and DHT-treated PCOS mouse models showed continuous anestrous cycle, devoid of corpus luteum in the ovary and increase of atretic follicles having a cystic-like structure compared with vehicle mice. Conversely, 11KT-treated mice showed prolonged estrous cycles and the exit of corpus luteum (Figure 1C and Figure 3F). Body weight gain was increased in DHT-treated mice but decreased in 11KT-treated mice in comparison with that in vehicle mice (Figure 1E). Notably, only 11KT-treated mice showed an increase in the number of preantral follicles (Figure 3F), consistent with the distribution of the increased number of preantral and small antral follicles observed in the ovaries of women with PCOS [9]. Additionally, 11KT-treated mice showed increases in both atretic preantral and antral follicles, but the atretic antral follicles were less than those in T- and DHT-treated mice (Figure 3G). Thus, 11KT may exhibit milder characteristics of PCOS as a weaker androgen, causing relatively minimal increases in estrogen secretion (Figure 5G) and less damage on ovulation such as prolonged estrous cycles and oligo-ovulation when compared with T and DHT. Notably, our 11KT-induced PCOS model exhibited milder hyperandrogenic traits, more closely mirroring the spectrum of androgen excess observed in certain PCOS subtypes. Importantly, our findings suggested that while 11KT is a weaker androgen, it still has the capacity to induce key features of PCOS, highlighting its potential role in the heterogeneous manifestations of the syndrome. Although PCOS animal models are commonly induced by DHT rather than T, DHT has significantly lower serum concentrations than 11KT in adult women [15]. Given that different PCOS phenotypes may arise from distinct androgen excess profiles, the reliance on DHT-induced models alone may not fully capture the diversity of PCOS pathology observed in patients. Therefore, developing more PCOS models is needed to investigate the underlying mechanisms by which hyperandrogenism contributes to PCOS pathogenesis. Our study provided compelling evidence that an 11KT-induced PCOS model may better represent a subset of PCOS patients characterized by elevated 11-oxygenated androgens. This model may offer valuable biological and clinical insights into human reproductive disorders in women with PCOS exhibiting mild symptoms. It should be noted that while 11KT represents a milder androgen compared to classical androgens, it does not exclude the possibility that other 11-oxygenated androgens may also play a significant role in PCOS pathogenesis. In this context, 11KT represents only a portion of the broader effects induced by 11-oxygenated androgens. Incorporating the analysis of transcriptome alterations induced by excess androgens such as 11KT in future studies is crucial to understand the impacts of hyperandrogenism on the comprehensive regulation of genes during the processes of steroidogenesis, folliculogenesis, and ovulation in PCOS. The distinct molecular mechanisms underlying androgen action—especially the variations in AR signaling pathways and downstream biological effects between classical and 11-oxygenated androgens—remain to be fully characterized and warrant further investigation. Although 11KT-treated mice did not exhibit significant weight gain, this does not imply an absence of metabolic disruption. Dysregulation of lipid metabolism, altered adipose tissue function, and insulin resistance may still occur, and dedicated metabolic studies are necessary to evaluate these potential effects. Additionally, whether excess 11KT impacts oocyte quality and subsequently affects fertility remains an important unresolved question. Future studies addressing the direct influence of 11KT on oocyte competence and developmental potential would greatly enhance our understanding of its role in reproductive dysfunction. Overall, by diversifying androgen-driven PCOS models, it can enhance our understanding of the molecular and physiological mechanisms contributing to PCOS heterogeneity, ultimately improving patient outcomes.

## 5. Conclusions

Heterogeneous androgen excess and multifaceted endocrine disruptions in PCOS give rise to distinct phenotypic variants with varying clinical presentations. By comparing the PCOS mouse models generated in this study with the human PCOS characteristics (Table 2), we demonstrated that different androgens exert distinct effects on gene expression, folliculogenesis, and steroidogenesis, contributing to the diverse pathophysiological mechanisms of PCOS. Herein, we provided a potential explanation for the heterogeneous clinical manifestations of PCOS and highlight the necessity of expanding androgen-driven PCOS models to capture the full complexity of this syndrome. Developing such models, including the 11KT-induced PCOS mouse, is crucial for refining our understanding of PCOS pathophysiology and advancing personalized therapeutic strategies.

## Figures and Tables

**Figure 1 biomedicines-13-01077-f001:**
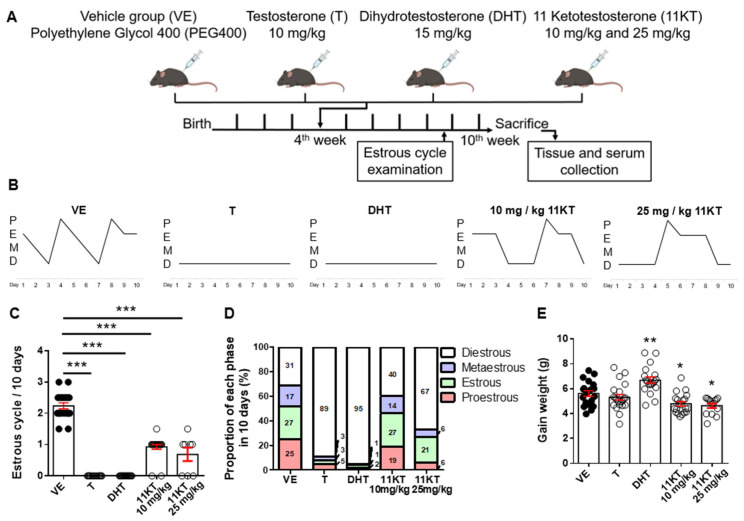
The pattern of estrous cycle in PCOS mouse models treated with different androgens. (**A**) Experimental design of PCOS mouse model. The 4-week-old C57BL/6 mice were injected with testosterone (T), dihydrotestosterone (DHT) or 11-ketotestosterone (11KT) by subcutaneous (s.c) for 6 weeks. Vehicle group mice were injected with PEG400. (**B**) 10-day estrous cycle pattern was showed in representative mice. P, proestrus; E, estrus; M, metestrus; D, diestrus. The difference in (**C**) cycle per 10 days, and (**D**) proportion of each estrous phase were shown. n = 8–15 per treatment group. (**E**) Weight gained from fourth week to tenth week was measured. n = 14–21 per treatment group. Data were analyzed by one-way ANOVA with the mean ± SEM; * *p* < 0.05, ** *p* < 0.01, *** *p* < 0.001.

**Figure 2 biomedicines-13-01077-f002:**
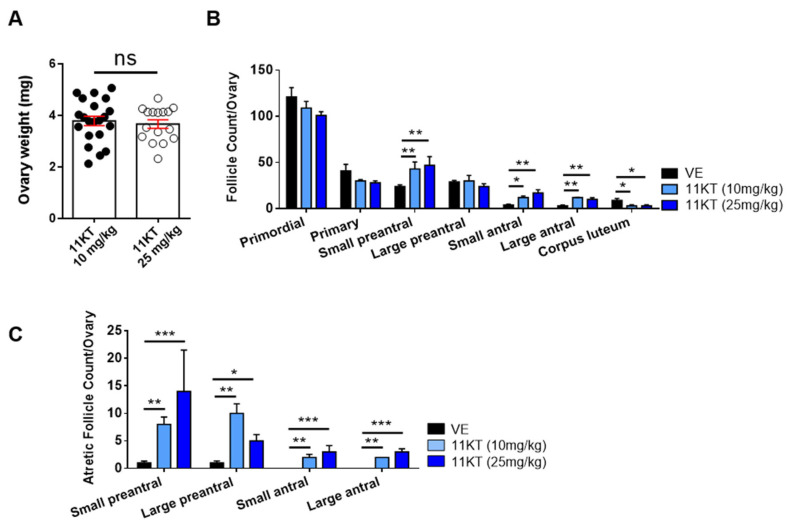
Follicle count of two dosages of 11KT-induced PCOS mouse models. (**A**) Ovary weight was compared in two dosages of 11KT-induced PCOS mouse models. (**B**) Histological sections of the number of follicles and (**C**) atretic follicles in 10 mg/kg (lower dose) and 25 mg/kg (higher dose) 11KT. Data of ovary weight were analyzed by Student’s *t*-test and follicle count were analyzed by one-way ANOVA with the mean ± SEM; ns, no significance; * *p* < 0.05; ** *p* < 0.01; *** *p* < 0.001.

**Figure 3 biomedicines-13-01077-f003:**
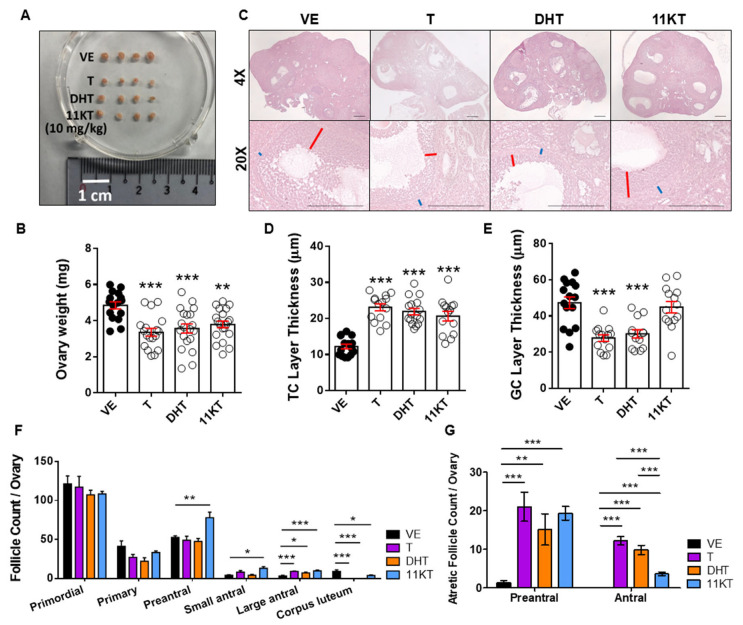
Comparison of ovary histomorphology and total follicle count in different stages of follicles from PCOS mouse models. (**A**) A representative photo showing the ovary size for each treatment group. (**B**) Ovary weight was measured. n = 16–21 per treatment group. (**C**) Representative ovary sections in each treatment groups. Bars represent 200 μm. 13–16 follicles for each treatment group were assess for (**D**) thickness of theca cells layer and (**E**) thickness of granulosa cells layer. Red lines represent granulosa cells layer thickness and blue lines represent theca cells layer thickness. (**F**) Histological ovarian sections of the number of follicles. (**G**) Histological sections of the number of atretic preantral and atretic antral follicle in each treatment group. VE, vehicle group; T, 10 mg/kg testosterone treatment group; DHT, 15 mg/kg DHT treatment group; 11KT, 10 mg/kg 11KT treatment group. Data were analyzed by one-way ANOVA with the mean ± SEM; * *p* < 0.05; ** *p* < 0.01; *** *p* < 0.001.

**Figure 4 biomedicines-13-01077-f004:**
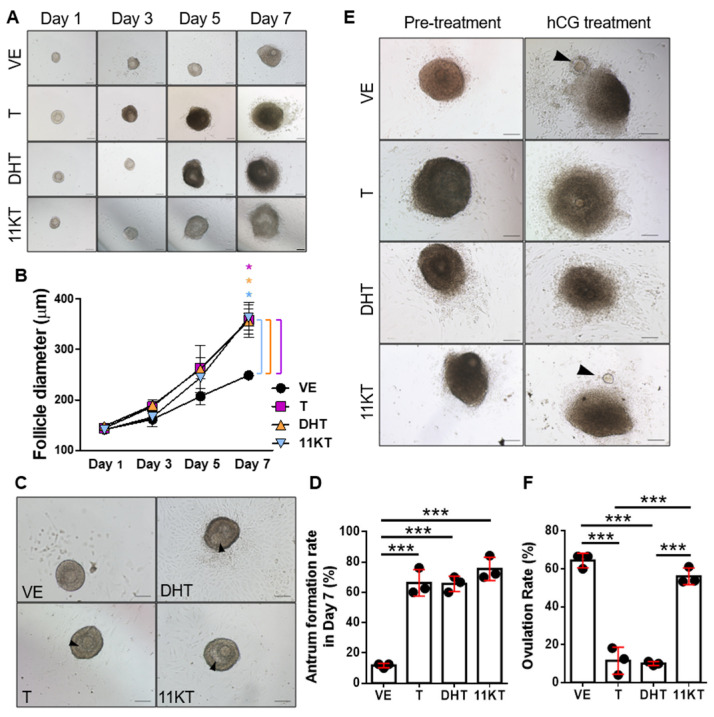
The effect of hyperandrogenism on follicle via in vitro follicle culture. (**A**) Morphology of preantral follicles cultured over 7 days. (**B**) Follicle growth in diameter was measured in day 1, 3, 5 and 7. (**C**) The structure of antrum (arrows) was observed in day 7 and (**D**) the antrum formation rates were calculated. An hCG-induced ovulation assay was performed after 6 days of preantral follicle culture. (**E**) Representative morphology of follicles pretreated with hCG and treated with hCG for 18 h. Arrows represent oocytes. (**F**) Ovulation rate upon hCG stimulation was calculated. At least 8 follicles per treatment group were performed in three independent experiments and data were analyzed by one-way ANOVA and are expressed as the mean ± SEM; * *p* < 0.05; *** *p* < 0.001. Bars represent 100 μm. VE, vehicle group (ethanol-treated).

**Figure 5 biomedicines-13-01077-f005:**
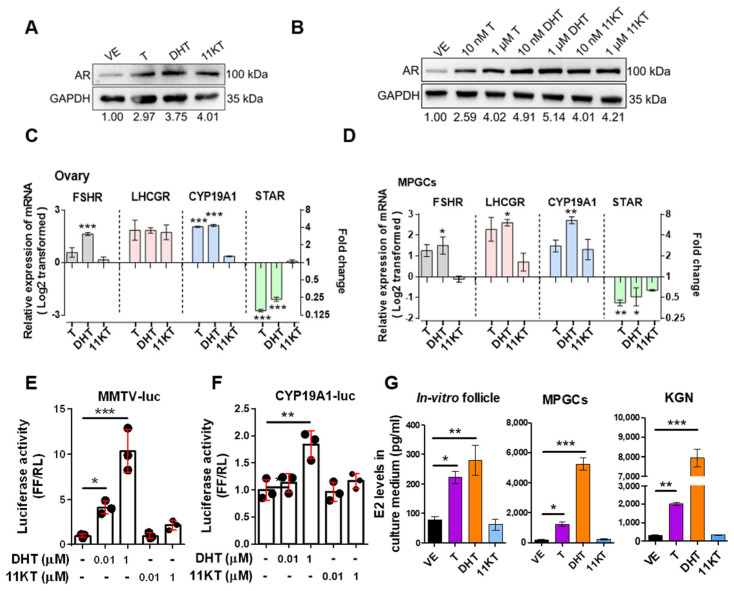
AR expression in the ovaries of PCOS mouse models and MPGCs with different androgen treatments. Western blotting of AR (**A**) in samples pooling three ovaries from hyperandrogenic mice and (**B**) in MPGCs treated with androgen for 24 h. Expression changes of FSHR and LHCGR (**C**) from samples pooling three ovaries from hyperandrogenic mice (n = 3) and (**D**) in MPGCs samples, each cultured for 72 h in the presence of testosterone, DHT, and 11KT (n = 3). KGN was transiently transfected with (**E**) the MMTV–luc and (**F**) *CYP19A1*-luc. Reni-luc (SV40) was used to control transfection efficiency. DHT or 11KT was added for 24 h and transcriptional function of AR was then quantified by the dual-luciferase assay after 48 h. The levels of E2 in (**G**) follicle, MPGCs and KGN culture medium were tested by ELISA after testosterone, dihydrotestosterone, and 11-ketotestosterone treatments (1 μM each) for 48 h (n = 3). Data were analyzed by one-way ANOVA and are expressed as the mean ± SEM; fold change indicated in right-hand *y*-axis; * *p* < 0.05; ** *p* < 0.01; *** *p* < 0.001. VE, vehicle group (ethanol-treated).

**Table 1 biomedicines-13-01077-t001:** Hormone levels in serum from androgenized mouse models.

	Vehicle Group	Androgenized Mouse Models				F Value	η^2^
		T	DHT	11KT (10 mg/kg)	11KT (25 mg/kg)		
LH (pg/mL)	60.40 ± 3.41	98.57 ± 13.23 ***	87.40 ± 13.87 **	90.73 ± 13.13 ***	95.07 ± 6.77 ***	12.0	0.64
T (pg/mL)	40.52 ± 24.46	244.09 ± 51.14 ***	31.96 ± 16.33	29.75 ± 2.56	15.11 ± 1.88	59.2	0.88
DHT (pg/mL)	38.15 ± 7.94	48.95 ± 17.05	492.24 ± 230.42 ***	58.92 ± 19.20	62.81 ± 18.68	21.5	0.77
11KT (pg/mL)	18.69 ± 1.99	30.02 ± 9.95	39.07 ± 3.61	3112.19 ± 684.94 ***	3167.67 ± 758.24 ***	104.6	0.93
E1(pg/mL)	1.29 ± 0.16	1.60 ± 0.18 *	1.59 ± 0.37 *	1.28 ± 0.19	1.12 ± 0.08	8.1	0.52
E2 (pg/mL)	39.17 ± 4.42	48.79 ± 8.45 *	49.77 ± 5.26 **	43.34 ± 4.59	44.27 ± 5.57	3.8	0.35

Data are the mean ± SEM (n = 5–9 per treatment group); Significant difference compared with vehicle group by one-way ANOVA; * *p* < 0.05; ** *p* < 0.01; *** *p* < 0.001.

**Table 2 biomedicines-13-01077-t002:** Table summary of Traits in human PCOS and hyper-androgenized mouse model in our study.

Human PCOS Traits	Mouse Models		
	T	DHT	11KT
Irregular Cycle (IC)/Acyclicity (A)	A	A	IC
Oligo (O)/Anovulation (A)	A	A	O
Hyperandrogenism	✓	✓	✓
↑ LH	✓	✓	✓
↑ Body weight *	✗	✓	✗
↑ Large antral follicles	✓	✓	✓
↑ Small antral follicles	✗	✗	✓
↑ Preantral follicles	✗	✗	✓
↑ Atretic antral Follicles	✓	✓	✓ **
↓ Granulosa cell layer thickness	✓	✓	✗
↑ Theca cell layer thickness	✓	✓	✓

✓, human PCOS trait present; ✗, human PCOS trait not present; * decrease in partial patients, ** less than T and DHT treated group.

## Data Availability

The original contributions presented in this study are included in the article. Further inquiries can be directed to the corresponding author.

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
