# Peer review of "Differential Effects of Canonical Androgens and 11-Ketotestosterone on Reproductive Phenotypes and Folliculogenesis in Mouse Model of PCOS"

_biomedicines, 2025, doi:10.3390/biomedicines13051077_

Round 1

Reviewer 1 Report

Comments and Suggestions for Authors

Differential Effects of Classical Androgens and 11-Ketotestosterone on Reproductive Phenotypes and Folliculogenesis in Mouse Models of PCOS

I have read the manuscript with interest, and you can find my appraisal as follows:

Abstract: The abstract is well-structured, and it contains relevant information. Despite this, I advise correcting it on the basis of the appraisal or raised concerns.

Introduction: The introduction is well-written, clear, simple, and informative. However, you need to add more information about the phenotypes of PCOS at the start of the section. Moreover, I have the link between hyperandrogenism and metabolic alterations in PCOS less clear. In this way, I suggest adding more information about. By the way, the link between human PCOS and the mouse model is well delineated. This is important since hirsutism is quite difficult to assess in animal models, but it is important for human PCOS. The hypotheses at the end of the section are in line with the information given in the introduction.

Methods: The methods are well written and informative, and the detailed information given allows the replication of the study. However, please, add more information about the plugin of ImageJ that you used. In the statistical analysis, you need to add the criteria you used relevant for the application of the parametric tests (normality, homoscedasticity, skewness, and kurtosis).

Results: I have appreciated that you have summarized the scope of the study and the phases. This is helpful for the reading. Table 1 needs to be improved, since it is not clear the test you have performed (t or ANOVA) and the corresponding values of F and t. Moreover, I suggest adding the Eta sqr (effect size) where appropriate. These values need to be reported in the section, as required for Table 1. Please, add them to the section. Moreover, please check the quality of some of the graphs that you reported (it is only a stylistic issue).

Discussion: In the discussion, I suggest adding a paragraph about “our 11KT-induced PCOS model exhibited milder hyperandrogenic traits, more closely mirroring the spectrum of androgen excess observed in certain PCOS subtypes.” This statement is in the conclusion, but I suggest moving it to the discussion. Moreover, the limitations of your study need to be stated better, as well as the future directions, which are better located in the discussion.

Author Response

Comment for Abstract: The abstract is well-structured, and it contains relevant information. Despite this, I advise correcting it on the basis of the appraisal or raised concerns.

Response for Abstract: Thank you for your comments and we highly appreciate your suggestions and have carefully revised the abstract to address the appraisal and concerns raised.

Comment for Introduction: The introduction is well-written, clear, simple, and informative. However, you need to add more information about the phenotypes of PCOS at the start of the section. Moreover, I have the link between hyperandrogenism and metabolic alterations in PCOS less clear. In this way, I suggest adding more information about. By the way, the link between human PCOS and the mouse model is well delineated. This is important since hirsutism is quite difficult to assess in animal models, but it is important for human PCOS. The hypotheses at the end of the section are in line with the information given in the introduction.

Response for Introduction: Thank you for your suggestion. We have added more information for the phenotypes of PCOS, and the link between hyperandrogenism and metabolic alterations in line 52 to 63.

Comment for Methods: The methods are well written and informative, and the detailed information given allows the replication of the study. However, please, add more information about the plugin of Image J that you used. In the statistical analysis, you need to add the criteria you used relevant for the application of the parametric tests (normality, homoscedasticity, skewness, and kurtosis).

Response for methods: Thank you for your suggestion. We utilized the basic functions of Image J for measurement of length, without the use of additional plugins in line 153-154. Additionally, we have revised the statistical analysis in line 214-217.

Comment for Results: I have appreciated that you have summarized the scope of the study and the phases. This is helpful for the reading. Table 1 needs to be improved, since it is not clear the test you have performed (t or ANOVA) and the corresponding values of F and t. Moreover, I suggest adding the Eta sqr (effect size) where appropriate. These values need to be reported in the section, as required for Table 1. Please, add them to the section. Moreover, please check the quality of some of the graphs that you reported (it is only a stylistic issue).

Response for Results: Thank you for improving our results in Table 1. We have revised the table 1 in line 260 and added the values of F and Æž² in the table

Comment for Discussion: In the discussion, I suggest adding a paragraph about “our 11KT-induced PCOS model exhibited milder hyperandrogenic traits, more closely mirroring the spectrum of androgen excess observed in certain PCOS subtypes.” This statement is in the conclusion, but I suggest moving it to the discussion. Moreover, the limitations of your study need to be stated better, as well as the future directions, which are better located in the discussion.

Response for Discussion: Thank you for your suggestion and comments. We totally agree with reviewer’s point of view, we have moved the sentence to line 406. Furthermore, we have added the limitations and future directions into the end of discussion (line 420-438).

Reviewer 2 Report

Comments and Suggestions for Authors

Reviewer’s Comments

General Comment

The authors have well designed the experiments and carried out without any flaws. The methodology section is well explained, Figure 1 is equal to a graphical abstract and neatly presented. The data analysis with statistics also appreciated.

There are very few suggestions and queries ,

  1. In the title authors have given “Classical Androgens” , what do they mean?
  2. In the title mouse models can be given as singular, mouse model.
  3. In the abstract lie no. 38 & 39, “. In vitro, all androgens enhanced follicle growth, but only T and DHT inhibited ovulation. How did you study inhibition of ovulation in vitro?
  4. From the line 73- 95, the term PCOS is used more than 20 times, is there any way you can modify the sentence?
  5. In the methodology section , authors can give Institutional animal ethical committee approval number for reference
  6. In the hormone assay, why authors chosen serum sample over plasma?
  7. In the line No. 198, “ELISAs for E2 quantification” but E2 is not explained before, hence the title can be ELISAs for 17β-estradiol (E2) quantification
  8. In Figure 3. (C), ovarian sections aree marked with red and blue lines, what are they?

Author Response

Comment 1: In the title authors have given “Classical Androgens”, what do they mean?

Response 1: Thank you for your question. We have revised “Classical Androgens” to "Canonical Androgens" which would be the better choice for a scientific paper title.

"Canonical" more precisely conveys that testosterone and DHT represent the standard, authoritative androgens against which others (like 11-ketotestosterone) are compared. The term effectively communicates their role as the established reference points in androgen research.

Comment 2: In the title mouse models can be given as singular, mouse model.

Response 2: Thank you for your suggestion. we have revised the title as reviewer suggested.

Comment 3: In the abstract line no. 38 & 39, “. In vitro, all androgens enhanced follicle growth, but only T and DHT inhibited ovulation. How did you study inhibition of ovulation in vitro?

Response 3: Thank you for your question. We utilized an in vitro follicle culture system to evaluate ovulatory potential by monitoring cumulus–oocyte complex (COC) expansion and oocyte release in response to ovulatory stimuli, such as hCG or EGF-like factors [1]. Inhibition of ovulation was assessed by the absence of COC expansion or oocyte extrusion, which are well-established indicators of ovulatory failure in cultured follicles, as observed under microscopic examination.

Comment 4: From the line 73- 95, the term PCOS is used more than 20 times, is there any way you can modify the sentence?

Response 4: Thank you for your question. We have revised this paragraph with improved flow and reduced repetition of the term.

Comment 5: In the methodology section, authors can give Institutional animal ethical committee approval number for reference

Response 5: Thank you for your question. We have mentioned approval number in ethical approval section. (line 480)

Comment 6: In the hormone assay, why authors chosen serum sample over plasma?

Response 6: Thank you for your question. Serum was chosen because it is the standard, widely accepted, and easily collected sample type for measuring steroid hormones [2].

Comment 7: In the line No. 198, “ELISAs for E2 quantification” but E2 is not explained before, hence the title can be ELISAs for 17β-estradiol (E2) quantification

Response 7: Thank you for your correction. we have revised. (line 206)

Comment 8: In Figure 3. (C), ovarian sections are marked with red and blue lines, what are they?

Response 8: Thank you for your question. We have written in figure legend 3: red lines represent granulosa cells layer thickness and blue lines represent theca cells layer thickness. (line 275)

  1. Chen, W., et al., Interlukin-22 improves ovarian function in polycystic ovary syndrome independent of metabolic regulation: a mouse-based experimental study. J Ovarian Res, 2024. 17(1): p. 100.
  2. Desai, R., D.T. Harwood, and D.J. Handelsman, Simultaneous measurement of 18 steroids in human and mouse serum by liquid chromatography-mass spectrometry without derivatization to profile the classical and alternate pathways of androgen synthesis and metabolism. Clin Mass Spectrom, 2019. 11: p. 42-51.

Round 2

Reviewer 1 Report

Comments and Suggestions for Authors

The manuscript has been improved as l required.

Author Response

Thanks for the reviewer's approval. The authors deeply appreciate your comments and suggestion.